# SYK Inhibition Potentiates the Effect of Chemotherapeutic Drugs on Neuroblastoma Cells In Vitro

**DOI:** 10.3390/cancers11020202

**Published:** 2019-02-10

**Authors:** Conny Tümmler, Gianina Dumitriu, Malin Wickström, Peter Coopman, Andrey Valkov, Per Kogner, John Inge Johnsen, Ugo Moens, Baldur Sveinbjörnsson

**Affiliations:** 1Molecular Inflammation Research Group, Department of Medical Biology, Faculty of Health Sciences, UiT The Arctic University of Norway, Hansine Hansens veg 18, 9019 Tromsø, Norway; gianina.dumitriu@uit.no (G.D.); ugo.moens@uit.no (U.M.); baldur.sveinbjornsson@uit.no (B.S.); 2Childhood Cancer Research Unit, Department of Women’s and Children’s Health, Karolinska Institutet, Tomtebodav 18A, 17177 Stockholm, Sweden; Malin.Wickstrom@ki.se (M.W.); Per.Kogner@ki.se (P.K.); John.Inge.Johnsen@ki.se (J.I.J.); 3IRCM, Inserm U1194, Université Montpellier, ICM, Institut régional du Cancer Montpellier, Campus Val d’Aurelle, 208 Rue des Apothicaires, 34298 Montpellier CEDEX 5, France; peter.coopman@inserm.fr; 4Department of Clinical Pathology, University Hospital of Northern Norway, Sykehusveien 38, 9019 Tromsø, Norway; andrej.yurjevic.valkov@unn.no

**Keywords:** pediatric cancer, neuroblastoma, tyrosine kinase, combination therapy

## Abstract

Neuroblastoma is a malignancy arising from the developing sympathetic nervous system and the most common and deadly cancer of infancy. New therapies are needed to improve the prognosis for high-risk patients and to reduce toxicity and late effects. Spleen tyrosine kinase (SYK) has previously been identified as a promising drug target in various inflammatory diseases and cancers but has so far not been extensively studied as a potential therapeutic target in neuroblastoma. In this study, we observed elevated *SYK* gene expression in neuroblastoma compared to neural crest and benign neurofibroma. While SYK protein was detected in the majority of examined neuroblastoma tissues it was less frequently observed in neuroblastoma cell lines. Depletion of SYK by siRNA and the use of small molecule SYK inhibitors significantly reduced the cell viability of neuroblastoma cell lines expressing SYK protein. Moreover, SYK inhibition decreased ERK1/2 and Akt phosphorylation. The SYK inhibitor BAY 61-3606 enhanced the effect of different chemotherapeutic drugs. Transient expression of a constitutive active SYK variant increased the viability of neuroblastoma cells independent of endogenous SYK levels. Collectively, our findings suggest that targeting SYK in combination with conventional chemotherapy should be further evaluated as a treatment option in neuroblastoma.

## 1. Introduction

Tyrosine kinases are important mediators of cellular functions such as proliferation, differentiation, metabolism, and survival. As tyrosine kinases are frequently deregulated in e.g. inflammatory diseases and cancer they are among the most attractive drug targets [1,2,3]. Spleen tyrosine kinase (SYK) is a 72 kDa non-receptor tyrosine kinase consisting of two SRC homology 2 (SH2) domains and a kinase domain joined by two linker regions, interdomains A and B [4]. A shorter SYK splice variant lacking 23 amino acids in the linker region B (SYK(S) or SYK B) has also been described in various cell types [5,6,7].

SYK is a multifunctional protein. It mediates inflammatory responses by linking immune cell receptors to various intracellular signaling networks and exhibits, for example, a pivotal role in B-cell development [4,8,9]. Among its various functions, SYK promotes, in concert with PKCδ, the expression of anti-apoptotic Mcl-1 in B-cell chronic lymphocytic leukemia (CLL) [10] and regulates actin filament assembly and dynamics through phosphorylation of cortactin and cofilin in ovarian cancer, thereby promoting migration and invasion [11]. Furthermore, phosphorylated SYK has been observed in specific cell types and areas of the developing nervous system and diverse functions have been described [12,13,14].

Widely expressed in hematopoietic cells [15,16], SYK is a promising therapeutic target in inflammatory diseases (including rheumatoid arthritis, allergies, systemic lupus erythematosus, and chronic immune thrombocytopenia) [17,18,19] as well as in different hematological malignancies such as CLL [10,20,21], non-Hodgkin lymphoma [22], and acute myeloid leukemia (AML) [23]. 

However, SYK expression is not restricted to hematopoietic cells and its presence has been described, among others, in epithelial, endothelial, and neuronal cells as well as in different solid tumors [24,25]. The role of SYK in the tumorigenesis of both hematological and solid malignancies is highly complex, tumor-specific and cell type-dependent, as both tumor promoting and suppressing functions have been described [25]. 

A tumor-suppressing role for SYK has been demonstrated in e.g. breast cancer [26], pancreatic cancer [27], melanoma [28], hepatocellular carcinoma [29] as well as childhood pro-B-ALL [30]. In contrast, SYK has been shown to be pro-tumorigenic in prostate cancer [31], small-cell lung cancer [32], ovarian cancer [11,33], glioma [34], pediatric retinoblastoma [35], and Ewing sarcoma [36].

Of note, distinct functions of the two SYK splice variants have also been demonstrated. While the longer SYK variant suppressed breast cancer invasiveness, the shorter variant SYK B did not [37].

Neuroblastoma is a malignancy of early childhood with 90% of patients diagnosed below the age of 10 [38]. Worldwide, neuroblastoma accounts for 12.5% of cancer cases in children of age 0 to 4 years [39]. Arising from the developing sympathetic nervous system, the majority of primary tumors occur in the adrenal gland [38]. About 50% of neuroblastomas are classified as low- and very-low-risk with a very good prognosis and receive minimum therapy [38]. In contrast, the treatment of high-risk neuroblastomas remains a challenge. Despite intensive therapy, about 50% of patients are refractory to first-line treatment or relapse within two years [40].

With the increasing understanding of neuroblastoma biology and the identification of druggable protein kinase targets such as ALK and Aurora A kinase as well as the MAPK and PI3K/mTOR/Akt signaling pathways targeted therapies, both alone and in combination with conventional drugs, provide new promising treatment options [41]. The aim of the present study was to investigate the expression of SYK in neuroblastoma tumor tissues as well as neuroblastoma cell lines and to evaluate its use as a potential therapeutic target. 

## 2. Results

### 2.1. SYK Is Expressed in Neuroblastoma Tissue

We examined *SYK* gene expression using the publicly available R2: Genomics analysis and visualization platform (http://r2.amc.nl) and observed that *SYK* expression was higher in four different neuroblastoma cohorts compared to neural crest cells and benign neurofibroma (Figure 1A).

Furthermore, we evaluated the presence of SYK protein in neuroblastoma and ganglioneuroma using immunohistochemistry (IHC). Appendix A displays the clinical features of the neuroblastoma tumors used in this study. SYK was present at varying levels in 40 out of 42 neuroblastomas and 3 out of 3 ganglioneuromas (Table 1). Figure 1B,C display a representative staining of SYK in non-*MYCN*-amplified and *MYCN*-amplified tumors, respectively. In both, the cytoplasm and the nucleus, a positive SYK staining was observed. SYK-positive tumor cells were present in 31 out of 32 non-*MYCN*-amplified neuroblastomas and in 9 out of 10 *MYCN*-amplified tumors (Table 1).

Using Fisher’s exact test we determined that there was no significant difference in the presence of SYK protein between *MYCN*-amplified and non-amplified tumors (*p* = 0.4239). However, examining different neuroblastoma datasets in the R2: Genomics analysis and visualization platform, we observed a significant negative correlation between *MYCN* and *SYK* expression (Appendix A displaying a representative dataset). In contrast, we found a significant positive correlation between *SYK* and *MYC* expression (Appendix A). Furthermore, we evaluated whether there was a difference in the presence of SYK in tumors that were treated with chemotherapy prior to surgery compared to untreated tumors. All 26 untreated tumor samples and 11 out of 13 treated tumor samples were SYK-positive. This difference was however not significant (Fisher’s exact test *p* = 0.1053). Of note, surgery was performed after at least 10–14 days of washout. Hence, no acute chemotherapy-induced regulation of genes should be expected.

Additionally, the presence of SYK phosphorylated at Tyr525, located within the activation loop of the kinase domain, was examined as an indication for active SYK [8,42]. Figure 1D,E display a representative staining of p-SYK in non-*MYCN*-amplified and *MYCN*-amplified tumors. A strong nuclear staining, as well as some cytoplasmic staining, was observed. Phospho-SYK was present in 29 out of 31 non-*MYCN*-amplified tumors, 9 out of 9 *MYCN*-amplified tumors, 25 out of 26 untreated tumors, and 10 out of 11 treated tumors (Table 1). 

Examples for SYK- and p-SYK-negative tumors are displayed in Appendix A. To ensure the specificity of the labeling, a corresponding isotype control antibody was used instead of the primary antibodies with which no apparent staining was observed (Appendix A).

### 2.2. SYK mRNA and to a Lesser Extend SYK Protein Are Present in Neuroblastoma Cell Lines

Using RT-PCR, western blot and immunocytochemistry (ICC) we examined the presence of *SYK* mRNA and protein in neuroblastoma cell lines. The majority of the neuroblastoma cell lines express *SYK* mRNA at varying levels (Figure 2A). However, SYK protein was detected by western blotting in only two of 10 neuroblastoma cell lines, even after long exposure times (Figure 2B). Interestingly, we noticed that the cell lines with absent or very low *SYK* mRNA levels are *MYCN*-amplified cell lines (SK-N-BE(2), SK-N-DZ, Kelly and IMR-32).

The shorter SYK splice variant SYK B has previously been detected in different cell types [5,6,7,37]. We observed that SH-SY5Y, LAN-6 and SK-N-FI cells concomitantly express both splice variants of *SYK* mRNA at similar levels whereas SH-EP1, SK-N-SH, and IMR-32 exhibit predominantly the short SYK B variant. The monocytic cell lines U937 and THP-1 with known SYK expression were used as positive controls for RT-PCR and western blot, respectively [43]. 

ICC was used to confirm the presence of SYK protein in SH-SY5Y and LAN-6 cells. A clear SYK labeling was observed in the cytoplasm of SH-SY5Y (Figure 2C) and LAN-6 cells (Figure 2D). The SYK signal appears to be localized mainly in the cytoplasm, with an increased intensity in patch-like structures. However, a faint staining was also observed in SK-N-BE(2) cells (Figure 2E). This could most likely be attributed to some moderate non-specific binding of the antibody. No staining was apparent in cells incubated with an isotype control antibody (Figure 2F–H). 

### 2.3. SYK Is Phosphorylated in Neuroblastoma Cell Lines

SYK activity is tightly controlled by its (auto)phosphorylation and important regulatory functions are associated with particular tyrosine residues, such as tyrosine 323, 352, 525, and 526 [8]. To determine whether SYK exhibits basic activity in neuroblastoma cell lines, we performed SYK immunoprecipitation followed by western blot with different phosphotyrosine-specific SYK antibodies in SH-SY5Y cell lysates and THP-1 cells, serving as a positive control. We detected phosphorylation of tyrosine 352 and tyrosine 323 in both cell types (Figure 3A). 

The presence of phosphorylated tyrosine 525/526 was examined by immunocytochemistry. Phospho-525/526 SYK was detected in the cytoplasm and close to the cell membrane in both SH-SY5Y cells (Figure 3B) and LAN-6 cells (Figure 3C). A weak nuclear staining was also noticeable. Variations in staining intensity were observed among the cells. As no prior synchronization or stimulation of the cells was performed, this may be attributed to the dynamic regulation of SYK-phosphorylation [8]. A very weak staining, most probably caused by non-specific background binding of the antibody, was observed in SK-N-BE(2) cells (Figure 3D) with no apparent SYK expression (Figure 2A,B).

### 2.4. Downregulation of SYK Reduces the Cell Viability of SYK Expressing Neuroblastoma Cell Lines

Using siRNA, we assessed the consequences of SYK knockdown on the cell viability of SYK-positive SH-SY5Y and LAN-6 as well as SK-N-BE(2) cells that show no apparent SYK expression. In SH-SY5Y and LAN-6, but not SK-N-BE(2) cells, we observed a significant decrease in cell viability 72 h post-transfection with two different SYK targeting siRNAs compared to scramble control siRNA (scr.) (Figure 4A). The decreased cell viability corresponded to reduced SYK protein levels in SH-SY5Y and LAN-6 cells (Figure 4B). Using densitometry, we calculated the downregulation of SYK protein. In SH-SY5Y cells the downregulation was approximately 5-fold with 18.1% (siRNA 1) and 21.3% (siRNA 2) of SYK protein remaining compared to the scrambled siRNA control (=100%). In LAN-6 cell the downregulation was less effective with 45.2% (siRNA 1) and 40.7% (siRNA 2) remaining SYK protein compared to the scrambled siRNA control.

### 2.5. SYK Activity Inhibition Decreases the Cell Viability of Neuroblastoma Cells

Four commercially available, pharmacological SYK inhibitors were used to evaluate the effect of SYK catalytic inhibition on the cell viability of SYK-positive SH-SY5Y and SYK-negative SK-N-BE(2) cells. Figure 5 displays the cell viability (MTT assay) after 48 h incubation with the SYK inhibitors BAY 61-3606 (Figure 5A), R406 (Figure 5B), PRT062607 (=P505-15; Figure 5C) and GS-9973 (=entospletinib; Figure 5D). The results of 24 h incubation with these inhibitors are shown in Appendix A and comparable tendencies could be observed. A statistically significant difference (control vs. treatment) in the cell viability of SH-SY5Y and SK-N-BE(2) cells was observed upon exposure to multiple inhibitor concentrations (marked by asterisks in the graphs). SH-SY5Y cells expressing high SYK levels were significantly more sensitive to the SYK inhibitors in comparison to SK-N-BE(2) cells expressing very low or no SYK (Figure 5). All four inhibitors significantly reduced the cell viability of both cell lines in a dose-dependent matter suggesting that at higher inhibitor concentrations the decrease in cell viability of SK-N-BE(2) is SYK-independent and caused by off-target effects.

Considering only inhibitor concentrations that impaired the cell viability of SH-SY5Y but not SK-N-BE(2) cells after 48 h of treatment, the concentration ranges were more narrow: 0.01–0.6 μM for BAY 61-3606, 0.8–1 μM for R406, and 1 μM for both PRT062607 and GS-9973. Because BAY 61-3606 displayed the most prominent differences between the two cell lines, it was used in subsequent experiments. 

### 2.6. Inhibition of SYK Activity Reduces ERK1/2 and Akt Phosphorylation in Neuroblastoma Cells

Active SYK is known to affect various downstream targets including MAPK, PI3K/Akt and NFκB signaling pathways [4,8,36]. Hence, we evaluated the consequences of SYK inhibition on ERK1/2 and Akt phosphorylation under the same experimental conditions used for the cell viability studies. As compared to vehicle alone, a significant and lasting reduction of ERK1/2 phosphorylation (Figure 6A,B) and Akt phosphorylation (Figure 6A,C) was observed for three out of four inhibitors (BAY 61-3606, R406, and PRT062607) after 4 h and 24 h. In contrast, GS-9973 treatment did not affect ERK1/2 or Akt phosphorylation at the investigated time points. 

### 2.7. The Selective SYK Inhibitor BAY 61-3606 Enhances the Effect of Chemotherapeutic Drugs on Neuroblastoma Cells

To determine whether SYK inhibition could increase the efficacy of chemotherapeutic agents to inhibit neuroblastoma cell growth, we combined BAY 61-3606 with the drugs paclitaxel, cisplatin, doxorubicin, and temozolomide, respectively. We investigated the effect of single vs. combined treatment on PARP cleavage (indicating apoptosis) and cell viability (MTT assay) in SYK-positive SH-SY5Y and SYK-negative SK-N-BE(2) cells (Figure 7A and Table 2). After 24 h, we observed an increase in cleaved PARP in SH-SY5Y cells for the combination of 0.4 μM BAY 61-3606 with paclitaxel, cisplatin, and temozolomide compared to single drug treatment (Figure 7A). For the combination of 0.4 μM BAY 61-3606 with paclitaxel or cisplatin this effect was sustained after 48 h (Appendix A).

Of note, after 24 h incubation with BAY 61-3606, paclitaxel or the combination temozolomide-BAY 61-3606 increased SYK protein levels could be observed (Figure 7A). This effect was not sustained after 48 h (Appendix A). However, after 48 h a clear reduction in SYK protein levels could be detected in cells treated with a combination of BAY 61-3606 and chemotherapeutic drugs compared to treatment with drugs alone (Appendix A). 

After 48 and 72 h incubation, a significant decrease in cell viability was evident for all drugs in combination with BAY 61-3606 in SH-SY5Y cells as compared to the chemotherapeutic drugs alone (Table 2).

In SK-N-BE(2) cells the apparent increase in cleaved PARP after combined treatment with BAY 61-3606 and paclitaxel was accompanied by a decrease in cell viability after 48 h and 72 h (Table 2). Comparing the combinations to treatment with BAY 61-3606 alone, a significant difference was observed for paclitaxel and cisplatin but not doxorubicin and temozolomide after 48 h and 72 h in SH-SY5Y cells, indicating that monotherapy with 0.4 μM BAY 61-3606 is comparable to a combination of temozolomide or doxorubicin and BAY 61-3606. 

Using Bliss independence and response additivity calculations, we determined whether the chemotherapeutic drug-BAY 61-3606 combination was synergistic, additive or antagonistic in regard to cell viability. The combinations of BAY 61-3606 and doxorubicin or temozolomide could not be analyzed with these methods due to the limited effects of these drugs as single agents at concentrations used in this study. A synergistic effect was determined for paclitaxel in combination with BAY 61-3606 in both SH-SY5Y and SK-N-BE(2) cells after 48 and 72 h (Figure 7B). Additionally, cisplatin in combination with BAY 61-3606 displayed a mainly additive effect in SH-SY5Y and SK-N-BE(2) cells after 48 and 72 h (Figure 7C).

A 24 h treatment with a higher BAY 61-3606 concentration (0.8 μM) in combination with any of the drugs resulted in a more pronounced increase in cleaved PARP in SH-SY5Y cells but not in the SYK-negative SK-N-BE(2) cells (Appendix A) after 24 h. However, after 48 h higher levels of PARP were observed in the combinations compared to treatment with chemotherapeutic drugs alone for the majority of drugs in SH-SY5Y and to a lesser extent in SK-N-BE(2) cells (Appendix A). Furthermore, a significant decrease in cell viability was observed for both cell lines comparing chemotherapeutic drug vs. combination with BAY 61-3606 after 48 and 72 h (Appendix A). However, when comparing the combinations to treatment with BAY 61-3606 alone, a significant difference in cell viability occurred for combinations of BAY 61-3606 with paclitaxel, cisplatin and temozolomide in SH-SY5Y cells after 48 h, and paclitaxel as well as cisplatin after 72 h. In SYK-negative SK-N-BE(2) cells combination of paclitaxel, cisplatin or doxorubicin and BAY 61-3606 after 72 h and paclitaxel as well as cisplatin after 48 h demonstrated a significant difference compared to monotherapy with BAY 61-3606.

A synergistic effect was determined for paclitaxel in combination with 0.8 μM BAY 61-3606 in both SH-SY5Y and SK-N-BE(2) cells after 48 and 72 h (Appendix A). Additionally, cisplatin in combination with 0.8 μM BAY 61-3606 displayed a less consistent effect, the combinational effects were mainly additive in SH-SY5Y and SK-N-BE(2) cells after 48 and 72 h (Appendix A) but some of the combinations were classified as synergistic or antagonistic using Bliss independence and response additivity calculations. 

### 2.8. Transfection with an Active SYK Variant Increases the Cell Viability of Neuroblastoma Cells Independent of Endogenous SYK Levels

To further explore the effect of SYK on neuroblastoma cell viability, we transfected SH-SY5Y, SK-N-BE(2), and SK-N-AS cells with expression vectors encoding different FLAG-tagged SYK variants (SYK wt = SYK wild type, SYK B = short SYK splice variant B, SYK Y130E = constitutive active SYK, SYK K402R = kinase dead SYK, SYK RR42/195KK = SYK with inactive SH2 domains). After 48 h, we measured the cell viability and confirmed transfection efficiency by evaluating exogenous SYK expression levels using western blot (Figure 8). SYK overexpression that exceeded the endogenous SYK levels was evident. Constitutive active SYK Y130E increased cell viability of all three neuroblastoma cell lines in comparison to transfection with the empty vector independent of the presence/absence of endogenous SYK (Figure 8A). Transfection with SYK wt significantly increased the cell viability of SK-N-AS cells and SYK RR42/195KK increased the cell viability of SH-SY5Y cells. A minor, reproducible reduction in cell viability after transfection with the kinase dead SYK mutant was observed in SH-SY5Y and SK-N-BE(2) cells, which was however not statistically significant.

Following transfection, all SYK variants were detected in the three cell lines by western blot although SYK RR42/195K was expressed at somewhat lower levels (Figure 8B).

## 3. Discussion

There is a need for further advancements in the treatment of neuroblastoma to improve the survival of high-risk patients and reduce acute and long-term toxic effects in neuroblastoma survivors. Targeted therapies exhibit great potential used either alone or more particularly in combination with conventional drugs [41,44]. 

In two other pediatric cancers, retinoblastoma and Ewing sarcoma, SYK promotes tumor cell survival and SYK inhibition, using small molecule inhibitors, was identified as a promising treatment option for these diseases [35,36].

In the present study, we observed that *SYK* expression was higher in four different neuroblastoma cohorts compared to neural crest cells and benign neurofibroma. Moreover, we demonstrate that SYK is present in neuroblastoma tissues and to a lesser extent in neuroblastoma cell lines. Inhibition of SYK using small molecule inhibitors alone or in combination with chemotherapeutic drugs as well as knockdown of SYK by siRNA impaired the cell viability of SYK expressing neuroblastoma cells. Additionally, SYK inhibition decreased phosphorylation of Akt and ERK1/2 indicating Akt and MAPK signaling as potential downstream targets of SYK in neuroblastoma. Furthermore, constitutive active SYK increased neuroblastoma cell viability independent of endogenous SYK expression. Taken together, our findings indicate a tumorigenic involvement of SYK in neuroblastoma.

We observed the presence of SYK protein in the majority of neuroblastoma tissues analyzed in this study. A positive staining of SYK and p-SYK was observed in both the cytoplasm and the nucleus with a more pronounced nuclear staining for p-SYK. The presence of SYK in different cellular compartments (cytoplasm, nucleus, membrane) has previously been described [37,45,46].

Furthermore, we compared the presence of SYK protein in *MYCN*-amplified and non-*MYCN*-amplified tumor tissue and did not observe differences between the two groups. Of note, by examining publicly available gene expression datasets, we detected a negative correlation between *MYCN* and *SYK* in neuroblastoma. In contrast, a positive correlation between *SYK* and *MYC* was discerned. *MYCN*-amplification occurs in about 20% of neuroblastomas and is associated with aggressive tumors and poor survival [47,48,49]. Furthermore, *MYC* has also been identified as an independent prognostic marker for poor survival in neuroblastoma [50] and is predominantly expressed by non-*MYCN*-amplified tumors [51]. A link between SYK and MYC has been previously demonstrated in Ewing sarcoma and hematopoietic cells [36,52]. Therefore, a potential connection between SYK and MYC in neuroblastoma is highly interesting and warrants further investigation. 

In contrast to neuroblastoma tumor tissue, SYK protein was only detected in two out of ten neuroblastoma cell lines by western blot. This is in accordance with previous findings by Alaminos et al. reporting more frequent methylation of the *SYK* promoter in neuroblastoma cell lines (60%) compared to tumor tissue (11%) [53]. In subsequent work by Margetts et al. and Grau et al. no aberrant hypermethylation of the *SYK* promoter was observed in tumor tissue or highly infiltrated bone marrow [54,55,56]. Yu et al. recently demonstrated that EGF stimulates SYK-mediated migration and invasion in ovarian cancer cells. The authors suggested that SYK function might be regulated by environmental stimuli [11]. Therefore, one could speculate that the absence of a specific stimulus, which is present in the tumor microenvironment, could lead to the downregulation of *SYK* expression in neuroblastoma cells lines. However, additional studies are necessary to determine if known factors, such as specific cytokines and chemokines, present in the neuroblastoma tumor microenvironment may affect SYK expression and function.

Complex phosphorylation events on tyrosine residues are required for the regulation of SYK functions by mediating conformational changes and creating docking stations for other proteins [8,57,58]. Phosphorylation of Tyr352 and/or Tyr348 (Tyr346 and Tyr342 in mouse SYK) provides binding sites for various proteins such as phospholipase Cγ, Vav-1 and 2 as well as Akt, and ERK, linking SYK to different signaling pathways and cellular functions [8,59,60,61,62]. Furthermore, Tyr352 has been linked to constitutive SYK activation [63]. The Tyr525/526 residues are located in the SYK activation loop. These phosphorylation sites likely provide important docking sites for other proteins thereby mediating intracellular signaling. However, mutations in these sites affect the in vitro catalytic activity only marginally [8,64,65]. Phosphorylation of Tyr323 in SYK promotes binding of Cbl protein family members causing ubiquitination and possibly degradation of SYK. However, Tyr323 is also an important binding site for PI3K indicating multiple functions of this phosphorylation site (reviewed in [8]). 

We determined the status of these three well-established SYK phosphorylation sites (Tyr352, Tyr525/526, and Tyr323) and found that all were phosphorylated under normal growth conditions in the SYK expressing SH-SY5Y neuroblastoma cells, suggesting the presence of catalytically active SYK. 

We observed a weaker nuclear staining of both total SYK and p-SYK in neuroblastoma cell lines as compared to neuroblastoma tissue. It has previously been demonstrated that SYK splice variants display differences in cellular localization and function [37,66,67] and that EGF can modulate SYK splicing pattern [66]. These findings indicate that the cellular localization of SYK may be affected by environmental stimuli and changes in splicing pattern. Therefore, the presence of specific stimuli in the neuroblastoma tumor microenvironment as well potential differences/changes in SYK splicing pattern compared to neuroblastoma cell lines may contribute to the observed differences in staining pattern. In our study, we observed that siRNA-mediated SYK downregulation reduced neuroblastoma cell growth. However, we did not achieve a complete SYK knockdown. Since SYK is a protein kinase, residual SYK protein could provide an explanation for the significant, but modest effect on the cell viability.

Therefore, we investigated the effects of commercially available SYK inhibitors BAY 61-3606 [68], R406 [69], PRT062607 (P505-15) [70] and GS-9973 [71] on neuroblastoma survival. Using increasing inhibitor concentrations, we compared the effect on cell viability in neuroblastoma cells with and without detectable SYK protein levels (SH-SY5Y and SK-N-BE(2), respectively). We determined at least one concentration for each inhibitor at which a significant reduction in cell viability was observed in SH-SY5Y but not SK-N-BE(2) cells, indicating an effect on cell viability that can likely be attributed to specific SYK inhibition.

PRT062607 and GS-9973 significantly reduced the viability of SH-SY5Y but not SK-N-BE(2) cells at a concentration of 1 μM. This is in line with recent work by Sun et al. where 1 μM PRT062607 and GS-9973 significantly impaired clonogenicity and cell viability of Ewing sarcoma cell lines [36]. A dose-dependent impairment of cell viability as well as increased caspase-3 activity has been previously reported in retinoblastoma cells after treatment with BAY 61-3606 and R406 [35]. We observed that these inhibitors significantly decreased SH-SY5Y cell viability as compared to SK-N-BE(2) cells at concentrations of 0.1–0.6 μM and 0.8–1 μM, respectively.

Of note, inhibitor concentrations >1 μM also significantly reduced the cell viability of SK-N-BE(2) cells that exhibit no apparent SYK expression suggesting off-target effects for all four inhibitors when used at higher concentrations. Kinase inhibitors commonly display off-target effects that can be beneficial but need to be carefully evaluated at the mechanistic level. For example, the SYK inhibitor BAY 61-3606 has been reported to inhibit JNK [72]. JNK inhibition has previously been demonstrated to reduce the apoptotic effect of chemotherapeutic drugs such as doxorubicin in SH-SY5Y cells [73]. Therefore, off-target effects of BAY 61-3606 on JNK could potentially impair doxorubicin function and not display a potentiating effect as seen in our studies. Furthermore, work by Colado et al. demonstrated that GS-9973 and R406 can impair T-cell function via off-target effects on the SYK homolog ZAP-70 [74]. We screened the neuroblastoma cell lines used in this study for ZAP-70 presence and found it expressed at protein level only in SK-N-DZ cells (Appendix A). Since this cell line was not used in the inhibitor studies, potential off-target effects on ZAP-70 can be excluded. However, these are just two of many potential proteins that may be the target of non-selective inhibition.

Various downstream targets for SYK have been described in health and disease [4,8]. ERK- and Akt-mediated signaling is known to be affected by SYK inhibition in CLL and Ewing sarcoma [20,36]. We observed that Akt and ERK1/2 phosphorylation was decreased by the SYK inhibitors BAY 61-3606, R406 and PRT062607, but not GS-9973. A possible explanation could be differences in the kinetics of GS-9973 mediated SYK inhibition. Since we only examined two time points, (4 h and 24 h) rapid and transient effects might not have been detected. PI3K/Akt- and MAPK-mediated signaling was previously shown to promote neuroblastoma tumorigenesis [75,76,77,78,79]. Therefore, a decrease in Akt and ERK1/2 phosphorylation and activity could contribute to the impaired cell viability.

Yu et al. demonstrated an increased expression of SYK in paclitaxel-resistant ovarian cancer cells and that paclitaxel in combination with the SYK inhibitor R406 increased apoptosis in vitro and impaired tumor growth in vivo [33]. Paclitaxel is a chemotherapeutic drug that is rarely used for the treatment of neuroblastoma. It was, however, included in this study to determine if the additive effect seen in ovarian cancer cells [33] could also be observed in neuroblastoma cell lines. We furthermore analyzed the effect of three drugs used in first-line treatment and/or refractory and relapsed neuroblastoma: cisplatin, doxorubicin, and temozolomide. 

We compared the cell viability of neuroblastoma cells treated with cytostatic drugs alone or in combination with the pharmacological SYK inhibitor BAY 61-3606. Synergetic and additive effects were observed in SYK expressing SH-SY5Y cells for paclitaxel- BAY 61-3606 and cisplatin- BAY 61-3606 combination, respectively. Furthermore, 0.4 μM BAY 61-3606 potentiated the effects of doxorubicin and temozolomide in SH-SY5Y cells. Interestingly, a synergetic effect of BAY 61-3606 in combination with paclitaxel was also observed in SK-N-BE(2), a cell line without significant expression of SYK protein. Although BAY 61-3606 concentrations applied in this study are rather low, off-target effects are likely the cause for the observed effect.

Furthermore, we observed increased amounts of cleaved PARP in the SYK-positive SH-SY5Y cells after treatment with BAY 61-3606 (0.4 and 0.8 μM) in combination with the tested drugs after 24 h, particularly when 0.8 μM BAY 61-3606 was used. We propose that the decrease in cell viability in SH-SY5Y cells may be attributed to an increase in apoptosis, whereas the decrease in cell viability in SYK-negative SK-N-BE(2) cells might rather be due to reduced proliferation than increased apoptosis. However, further experiments are required to determine the detailed mechanisms.

In addition, we also demonstrate that transient transfection with a constitutively active SYK variant increased the cell viability of neuroblastoma cell lines independent of endogenous SYK expression. This suggests that SYK has tumor-promoting functions in neuroblastoma. Previous work reported a tumor-suppressing role for SYK in breast cancer, among others. Transfection of breast cancer cells with a wild-type SYK encoding vector suppressed invasive outgrowth in Matrigel and impaired tumor growth and metastasis in mice [26]. We did, however, not observe any inhibitory effect on the cell viability of transfected neuroblastoma cell lines expressing exogenous SYK. Taken together, this suggests that SYK functions as a tumor-promoting molecule in neuroblastoma rather than having a tumor-suppressing effect. 

To further evaluate the potential therapeutic use of SYK inhibitors in neuroblastoma both as a single agent and in combination with existing chemotherapeutic drugs, in vivo studies are necessary. Since SYK is widely expressed by hematopoietic cells, potential negative effects on the immune cells of the tumor microenvironment have to be carefully evaluated using immunocompetent neuroblastoma animal models. Recent work in glioma demonstrated that SYK inhibition impaired the mobility and infiltration of B cells and CD11b+ leukocytes in addition to reducing proliferation and migration of tumor cells [34].

## 4. Materials and Methods

### 4.1. Microarray Gene Expression

Gene expression analysis was performed using the MegaSampler feature of the publicly available R2: Genomics Analysis and Visualization Platform (http://r2.amc.nl).

### 4.2. Reagents and Antibodies

The selective SYK inhibitors GS-9973 (Entospletinib) and R406 were purchased from Selleck Chemicals Europe (Munich, Germany). BAY 61-3606 and PRT062607 were obtained from Calbiochem/Merck (Merck Life Science AS, Oslo, Norway) and ApexBio (Houston, TX, USA), respectively. Paclitaxel, cisplatin, doxorubicin, and temozolomide were bought from Sigma-Aldrich Norway AS (Oslo, Norway). The antibodies used in this study are listed in Table 3.

### 4.3. Human Tissue Samples and Cell Lines

Human tissue samples were obtained, with informed consent (written or verbal) provided by the parents or guardians for the use of tumor samples in research, in accordance with the ethical approval from the Stockholm Regional Ethical Review Board and the Karolinska University Hospital Research Ethics Committee (approval ID 2009/1369-31/1 and 03-736). Neuroblastoma tumor tissue was collected at the Karolinska University Hospital, snap-frozen in liquid nitrogen and stored at −80 °C until further use.

The human cell lines SK-N-AS, SK-N-SH, SK-N-DZ, SK-N-FI, SH-EP1, Kelly, SH-SY5Y, and IMR-32 as well as THP-1, Jurkat E6.1, and U937 cells were obtained from the ATCC (American Type Culture Collection, LGC Standards GmbH, Wesel, Germany). SK-N-BE(2) cells were purchased from DSMZ (Deutsche Sammlung von Mikroorganismen und Zellkulturen, Braunschweig, Germany). The cell lines were cultivated in RPMI-1640 medium containing L-glutamine and sodium bicarbonate (Sigma-Aldrich Norway AS, Oslo, Norway) supplemented with 10% heat-inactivated fetal bovine serum (FBS; Thermo Fisher Scientific Inc., Waltham, MA, USA). LAN-6 cells were a kind gift from Deborah Tweddle, Newcastle University and were grown in Iscove’s Modified Dulbecco’s Medium (Sigma-Aldrich Norway AS, Oslo, Norway) supplemented with GlutaMAX™ and 10% heat-inactivated FBS. All cell lines were cultivated at 37 °C in humidified air with 5% CO_2_ and mycoplasma tests were performed regularly using the MycoAlert™ PLUS Mycoplasma Detection Kit (Lonza, Basel, Switzerland). The identity of the human neuroblastoma cell lines was confirmed by STR-profiling performed at the Centre of Forensic Genetics, University of Tromsø, Norway. 

### 4.4. Immunohistochemistry (IHC)

Formalin-fixed, paraffin-embedded tissue sections were deparaffinized using xylene (VWR International, Oslo, Norway) and a series of graded alcohols (Sigma-Aldrich Norway AS, Oslo, Norway) followed by rehydration and washing in phosphate buffered saline (PBS, Biochrom GmbH, Berlin, Germany). Antigen retrieval was performed in sodium citrate buffer (pH 6) in a microwave oven. After blocking of endogenous peroxidase with 0.3% H_2_O_2_ for 15 min, unspecific antibody binding sites were blocked with 5% BSA in PBS (AppliChem, Darmstadt, Germany) for 45 min. The sections were incubated with the primary antibody overnight at 4 °C. The following day sections were thoroughly washed in PBS and incubated with SignalStain® Boost IHC Detection Reagent, HRP, Rabbit (Cell Signaling Technology, Leiden, Netherlands) and kept for 1 h at room temperature. Following washes in PBS, the sections were incubated with Liquid DAB+ Substrate solution (Dako, Agilent Technologies, Inc., Santa Clara, CA, USA). A matched isotype control was used as a control for nonspecific staining. The sections were examined with a BX43 microscope (Olympus, Tokyo, Japan) and images were acquired with an Olympus DP26 camera.

### 4.5. RNA Isolation and RT-PCR

Total RNA was isolated using the NucleoSpin® TriPrep Kit (MACHEREY-NAGEL GmbH & Co. KG, Düren, Germany) and RNA quantity and quality was determined with a NanoDrop™ 2000 spectrometer (Thermo Fisher Scientific Inc.). One μg RNA was used as input for cDNA synthesis with the iScript™ cDNA Synthesis Kit (Bio-Rad Laboratories AB, Oslo, Norway).

PCR was set up as a 25 μl reaction mix containing 2 μL cDNA, 12.5 μL AccuStart™ II GelTrack PCR SuperMix (Quanta Biosciences, Gaithersburg, MD, USA), 400 nM of each primer (Sigma-Aldrich Norway AS, Oslo, Norway) and 10.1 μL of ultra-pure H_2_O (Biochrom GmbH, Berlin, Germany). The PCR run was performed in a T100™ Thermal Cycler (Bio-Rad Laboratories AB) as follows: 2 min at 94 °C and 35 cycles of 94 °C for 20 s, 61 °C for 30 s and 72 °C for 90 s. The following intron spanning primer sets were used: APRT (housekeeping) 5′ CCCGAGGCTTCCTCTTTGGC 3′ (sense) and 5′ CTCCCTGCCCTTAAGCGAGG-3′ (antisense) [80], SYK 5′ CATGTCAAGGATAAGAACATCAT AGA 3′ (sense) and 5′ AGTTCACCACGTCATAGTAGTAATT 3′ (antisense) [26], SYK L/S 5′ TTTTGGAGGCCGTCCACAAC ‘3 (sense) and 5′ ATGGGTAGGGCTTCTCTCTG 3′ (antisense) [37].

All primer sets used in this study were intron-spanning to avoid false positive signals caused by amplification of residual traces of genomic DNA. PCR products were analyzed by gel electrophoresis. The 2% SeaKem® LE Agarose gel (Lonza) was stained with GelRed™ (Biotium, Inc., Hayward, CA, USA) and visualized under UV light in the BioDoc-It™ Imaging System (UVP, LLC, Upland, CA, USA).

### 4.6. Immunocytochemistry (ICC) and Western Blot

Cells were grown in 8-well µ-Slide dishes (iBidi GmbH, Munich, Germany) until they reached approximately 70% confluence. Following a brief rinse with PBS the cells were fixed with 4% formaldehyde (Alfa Aesar, VWR International, Oslo, Norway) for 20 min. After three washes with PBS, unspecific binding sites were blocked with 5% goat serum (Sigma-Aldrich Norway AS) in PBS containing 0.3% Triton-X-100 (Sigma-Aldrich Norway AS) for 1 h. The cells were incubated with primary antibodies diluted in 1% bovine serum albumin (BSA; AppliChem, Darmstadt, Germany) in PBS containing 0.3% Triton-X-100 overnight at 4 °C. After three washes with PBS, the cells were incubated with the secondary antibodies diluted in 1% bovine serum albumin in PBS containing 0.3% Triton-X-100 for 1 h at room temperature, protected from light. Following three washes with PBS, the nuclei were stained with Hoechst 33342 (ImmunoChemistry Technologies, LLC, Bloomington, IN, USA) for 10 min. The cells were washed 3x with PBS and covered with Mounting Medium for fluorescence microscopy (iBidi GmbH). Subsequently, the cells were examined using a Zeiss LSM780 confocal microscope (Carl Zeiss AG, Oberkochen, Germany). Images were taken with the same microscope settings (laser intensity, gain etc.) and identical image processing parameters were applied to allow comparison between the cell lines. Western blots were performed as previously described [81].

### 4.7. Immunoprecipitation (IP)

Cells were washed twice with cold PBS and lysed by addition of lysis buffer containing 50 mM Tris-HCl (pH 8.0), 150 mM NaCl, 0.1% Triton X-100, 1 mM DTT and 1 mM EDTA (Sigma-Aldrich Norway AS, Oslo, Norway) as well Halt™ Protease and Phosphatase Inhibitor Cocktail (Thermo Fisher Scientific Inc.). Following sonication and centrifugation, the protein concentration was determined using a Protein Quantification Assay (MACHEREY-NAGEL GmbH & Co. KG). A sample of the supernatant was taken as “input control”, supplemented with NuPAGE® LDS Sample Buffer (4X) (Thermo Fisher Scientific Inc.) as well as 100 mM DTT (AppliChem, Darmstadt, Germany) and incubated for 10 min at 70 °C. Cell lysate containing approximately 800 μg protein was pre-cleared with an irrelevant IgG2a antibody (same class as the SYK antibody used for IP) to reduce unspecific binding. Afterwards, the samples were incubated with the anti-SYK antibody (1 μg) rotating at 4 °C overnight followed by incubation with 80 μl of a 50% Sepharose G beads/lysis buffer solution (GE Healthcare, Oslo, Norway) for 1 h rotating at 4 °C. The beads were washed three times with lysis buffer and two times with 50 mM Tris-HCl, pH 8.0. Afterwards, the beads were incubated in sample buffer containing NuPAGE® LDS Sample Buffer (4×), ultrapure H_2_O as well as 100 mM DTT for 10 min at 70 °C. The samples were subsequently used for western blot analysis.

### 4.8. siRNA-Mediated SYK Silencing

Two pre-designed SYK siRNAs were used in this study: ID-s13679 (siRNA 1) and ID-s13681 (siRNA 2) as well as a scramble control #1 siRNA (cat# 4392420 and 4390843, Ambion, Thermo Fisher Scientific Inc.). Cells were seeded into 24- and 6-well plates for cell viability assay and western blot analysis, respectively. After 24 h, the cells were transfected with the different siRNAs using Lipofectamine RNAiMAX reagent (Thermo Fisher Scientific Inc.) according to the manufacturer’s specifications. Briefly, cells were incubated with 5 pmol (24-well plate) or 30 pmol siRNA (6-well plate) per well in Opti-MEM (Thermo Fisher Scientific Inc.) for 4 h followed by the removal of the transfection mix and addition of fresh Opti-MEM. After 72 h the cell viability was assessed and cells were harvested for western blot analysis. Densitometry was performed using Fiji software [82].

### 4.9. Cell Viability Assay

To assess the effect of the commercially available SYK inhibitors BAY 61-3606, R406, GS-9973 (Entospletinib) and PRT062607 alone as well as BAY61-3606 in combination with paclitaxel, cisplatin, doxorubicin and temozolomide on the cell viability of SH-SY5Y and SK-N-BE(2) neuroblastoma cells the colorimetric MTT (3-(4,5-dimethylthiazol-2-yl)-2,5-diphenyltetrazodium bromide)-assay was used [83]. The cells were seeded in 96-well plates in full growth media. As an exception, cells treated with GS-9973 and PRT062607 were seeded in Opti-MEM to reduce cell viability variations attributed to residual serum. After 24 h, the cells were washed once with Opti-MEM before incubation with SYK inhibitors alone for 24 and 48 h or a combination of chemotherapeutic drugs with BAY61-3606 for 48 and 72 h. Control cells received the corresponding drug vehicle at the highest concentration present in the drug-treated cells. BAY 61-3606 and doxorubicin were dissolved in water, R406, GS-9973, PRT062607, and temozolomide in DMSO, paclitaxel in ethanol, and cisplatin in 0.9% saline. After 24, 48 or 72 h the MTT solution (10 μL of 5 mg MTT, Sigma-Aldrich Norway AS, per ml phosphate buffered saline) was added to each well and incubated for additional 3 h. To facilitate formazan crystal solubilizing, 70 μL of the solution were carefully removed from each well, 100 μL isopropanol containing 0.04 M HCl were added and mixed thoroughly. In addition, the plates were placed on an orbital shaker for 1 h at room temperature. The absorbance was measured with a CLARIOstar plate reader (BMG LABTECH, Ortenberg, Germany) at 590 nm. The experiment was performed at least three times with at least three parallels per treatment. The cell viability was calculated as the ratio of the mean OD of treated cells over vehicle treated control cells (100% living cells). The cell viability assay for the siRNA and SYK plasmid studies were performed in 24-well plates. The amounts of MTT solution and acidic isopropanol were adjusted correspondingly.

### 4.10. Cell Signaling Study

To investigate the effect of commercially available SYK inhibitors on MAPK- and Akt- mediated signaling SH-SY5Y cells were seeded in 6-well plates in full growth medium. The next day, the cells were washed in Opti-MEM and treated with BAY 61-3606, R406, GS-9973 (entospletinib), PRT062607 or corresponding vehicle controls (water for BAY 61-3606 and DMSO for R406, GS-9973, and PRT062607) for 4 or 24 h. Following incubation, the cells were washed with PBS and harvested in RIPA Lysis and Extraction Buffer containing Halt™ Protease and Phosphatase Inhibitor Cocktail (Thermo Fisher Scientific Inc.) and analyzed by western blot. Densitometry was performed using Fiji software. Phosphorylated and total protein were normalized to their respective GAPDH loading control (pERK/GAPDH, ERK/GAPDH, pAkt/GAPDH, Akt/GAPDH). Ratios of pERK/ERK and pAkt/Akt were calculated using the normalized values. The respective vehicle control was set as 1 and the ratios were calculated.

### 4.11. Transfection with SYK Plasmids

The following, previously described FLAG-tagged plasmids were used in this study: pCAF1 empty expression vector, SYK wt = SYK wild type, SYK B = short SYK splice variant B, SYK Y130E = constitutive active SYK, SYK K402R = kinase dead SYK, SYK RR42/195KK = SYK with inactive SH2 domains [26,46]. Cells were seeded in 24- (cell viability assay) or 6-well plates (western blot) in full growth media. The following day the cells were transfected using jetPRIME® transfection reagent (Polyplus-transfection®, Illkirch, France) according to the provided manual (1 μg DNA per well in a 6-well plate and 0.25 μg DNA per well in a 24-well plate). After 5 h, the media containing the transfection mix was removed and fresh media was added. After 48 h, cell viability was determined and cells were harvested for western blot.

### 4.12. Drug Combination Analysis

To determine whether the chemotherapeutic drug-SYK inhibitor combinations displayed a synergistic or additive effect, highest single agent, response additivity and Bliss independence calculations were performed on the cell viability data as described in [84]. Highest single agent method proves the superiority of the drug combination compared to its single agents and was assessed with statistical testing (two-way ANOVA). Response additivity and the Bliss Independence model both compares the observed drug combination effect to the expected additive effect and thereby calculates a combination index. For Response additivity the expected additive effect is calculated as following: E(A) + E(B) and for the Bliss Independence: E(A) + E(B) − E(A)*E(B) where E is the effect produced by drug A and B. A combination index <0.9 was defined as synergistic, 0.9–1.1 as additive and >1.1 as antagonistic.

### 4.13. Statistical Analysis

GraphPad Prism software (versions 7 and 8, GraphPad Inc., San Diego, CA, USA) was used for statistical analysis and graph design. Fisher’s exact test was used to test the statistical significance of the association between two categories. Two-way ANOVAs and Dunnett or Bonferroni post-tests were applied to assess two independent variables (differences between cell lines and the effect of treatment). Two-tailed one sample t-tests were used for the statistical analysis of the cell signaling studies.

## 5. Conclusions

Our findings demonstrate the presence of functional SYK in neuroblastoma tissue as well as certain neuroblastoma cell lines and indicate that pharmacological SYK inhibition may be a potential therapeutic approach that can be used to support conventional chemotherapy in SYK-expressing neuroblastomas.

## Figures and Tables

**Figure 1 cancers-11-00202-f001:**
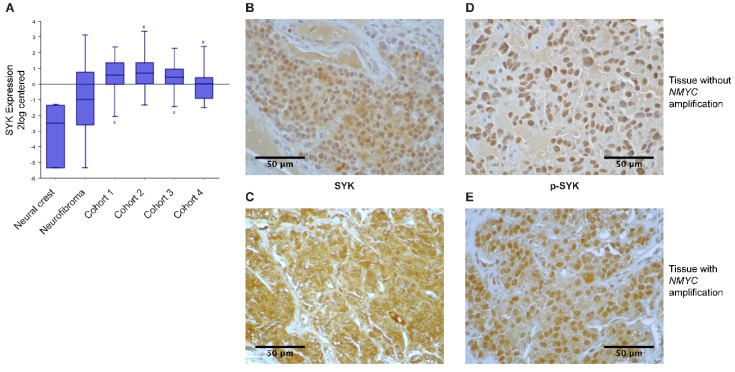
SYK is expressed in neuroblastoma tissue. Gene expression data were analyzed using the R2 database http://r2.amc.nl. (**A**) The expression of *SYK* was compared between neural crest (Etchevers n = 5), benign neurofibroma (Miller n = 86) and 4 neuroblastoma cohorts (cohort 1: Versteeg n = 88, cohort 2: Delattre n = 64, cohort 3: Hiyama n = 51, cohort 4: Lastowska n = 30). The presence of SYK protein (**B**,**C**) and phosphorylation at Tyr525 (**D**,**E**) were determined in neuroblastoma primary tissue using immunoperoxidase staining. (**B**,**D**) display a staining of a non-*MYCN*-amplified tumor and (**C**,**E**) show a *MYCN*-amplified tumor. Images were captured at a magnification of 900×. The displayed images are representative stainings from a panel of neuroblastoma tumors.

**Figure 2 cancers-11-00202-f002:**
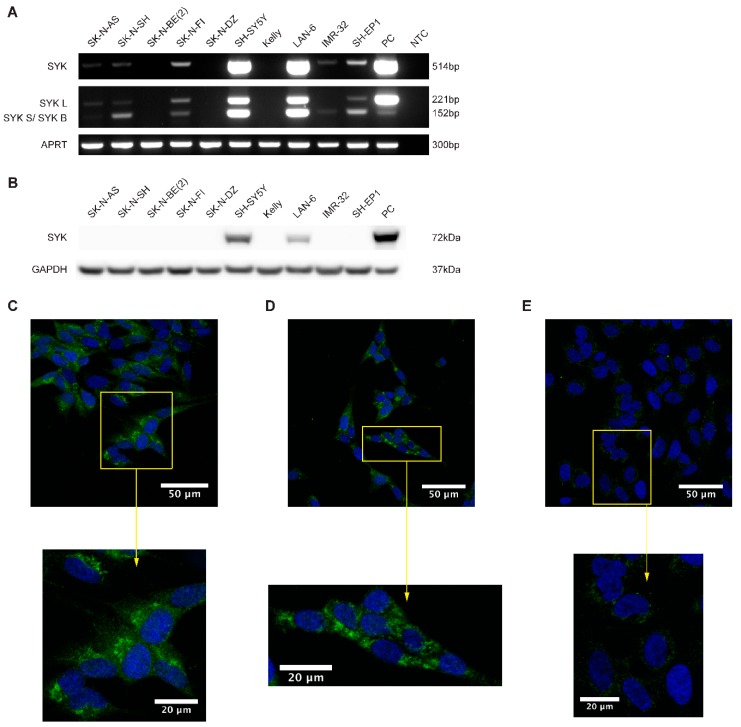
*SYK* mRNA and to a lesser extend SYK protein are expressed in neuroblastoma cell lines. (**A**) RT-PCR analysis demonstrating the expression of both *SYK* mRNA variants in different neuroblastoma cell lines. U937 cells were used as a positive control (PC). NTC, no template control. (**B**) Expression of SYK protein was determined by western blot. THP-1 cells were used as a positive control. Immunofluorescence labeling of SYK (green) in SH-SY5Y (**C**), LAN-6 (**D**) and SK-N-BE(2) cells (**E**). The nuclei (blue) were stained with Hoechst 33342. Panels (**F–H)** display isotype controls for SH-SY5Y (**F**), LAN-6 (**G**) and SK-N-BE(2) cells (**H**).

**Figure 3 cancers-11-00202-f003:**
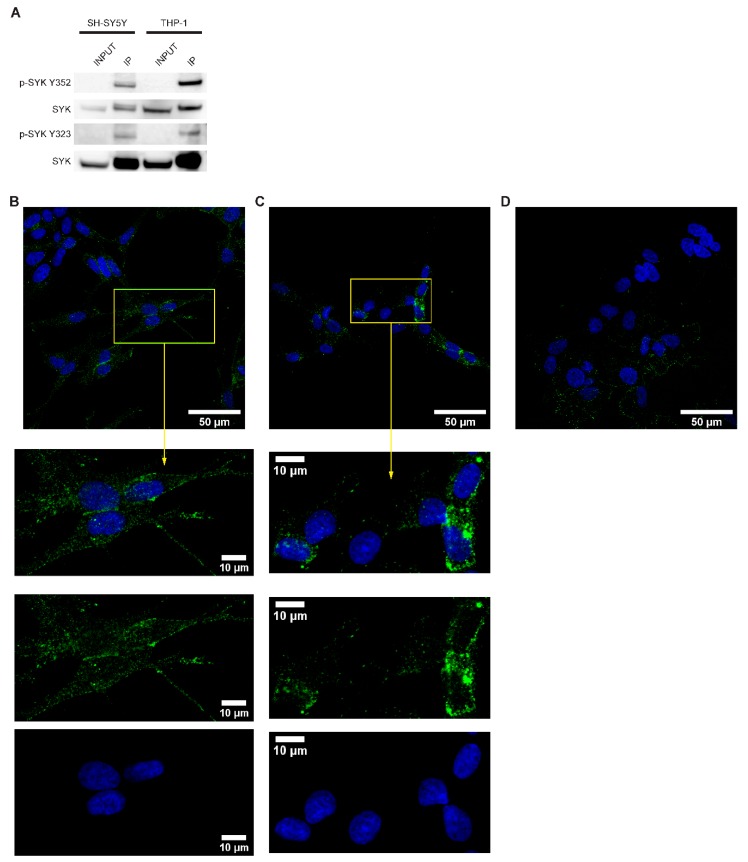
SYK is phosphorylated in neuroblastoma cell lines. (**A**) Immunoprecipitation with a SYK specific antibody was performed followed by western blot using antibodies against the SYK Tyr352 and Tyr323 phosphorylation residues. THP-1 cells were used as a positive control. Immunofluorescence labeling of p-SYK (Tyr525/526) in SH-SY5Y (**B**) and LAN-6 cells (**C**). SK-N-BE(2) cells (**D**) served as a negative control. The nuclei (blue) were stained with Hoechst 33342.

**Figure 4 cancers-11-00202-f004:**
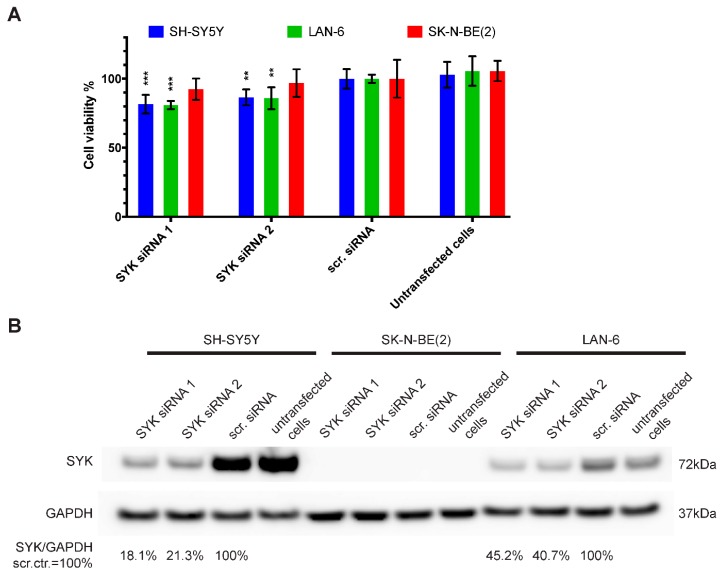
SYK downregulation reduces the cell viability of SYK expressing neuroblastoma cell lines. SYK-positive SH-SY5Y and LAN-6 cells, as well as SYK-negative SK-N-BE(2) cells, were transfected with 5 pmol SYK specific or scrambled (scr.) siRNA. The medium was replaced after 4 h. After 72 h, the cell viability was evaluated by MTT assay (**A**). The scr. control was set as 100% viable cells. Data are presented as mean ± SD from three independent experiments. Statistical comparisons were made using two-way ANOVA and a significant effect was observed for the siRNA treatment *p* < 0.001 and between cell lines *p* = 0.003. The Dunnett’s multiple comparison test was used to evaluate the difference between scr. siRNA vs. SYK specific siRNA: ** *p* < 0.01, *** *p* < 0.001. (**B**) SYK expression levels were determined by western blot.

**Figure 5 cancers-11-00202-f005:**
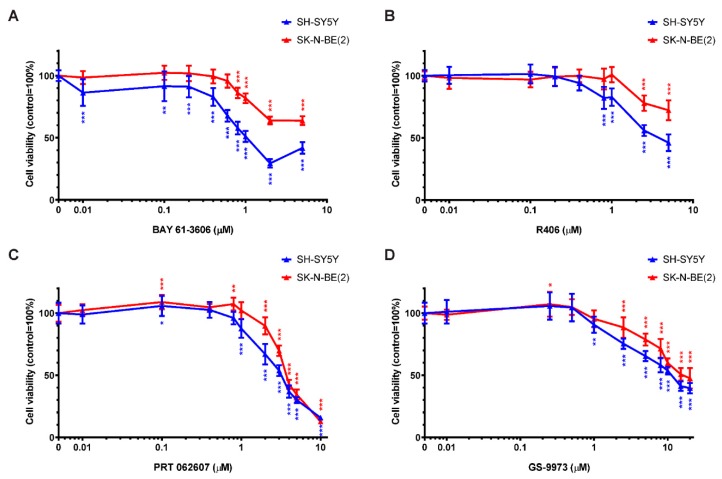
Inhibition of SYK decreases the cell viability of neuroblastoma cells. Cell viability was measured in SH-SY5Y and SK-N-BE(2) cells by MTT assay after 48 h incubation with increasing concentrations of the SYK inhibitors BAY 61-3606 (**A**) R406 (**B**) PRT062607 (**C**) GS-9973 (**D**). The control was set as 100% viable cells. Data are presented as mean ± SD from three independent experiments. Using two-way ANOVA, a significant difference between cell lines and a significant effect of the inhibitor *p* < 0.001 was seen. Dunnett’s multiple comparison test was used to evaluate the difference between vehicle treated control cells and the various inhibitor concentrations: * *p* < 0.05 ** *p* < 0.01 *** *p* < 0.001.

**Figure 6 cancers-11-00202-f006:**
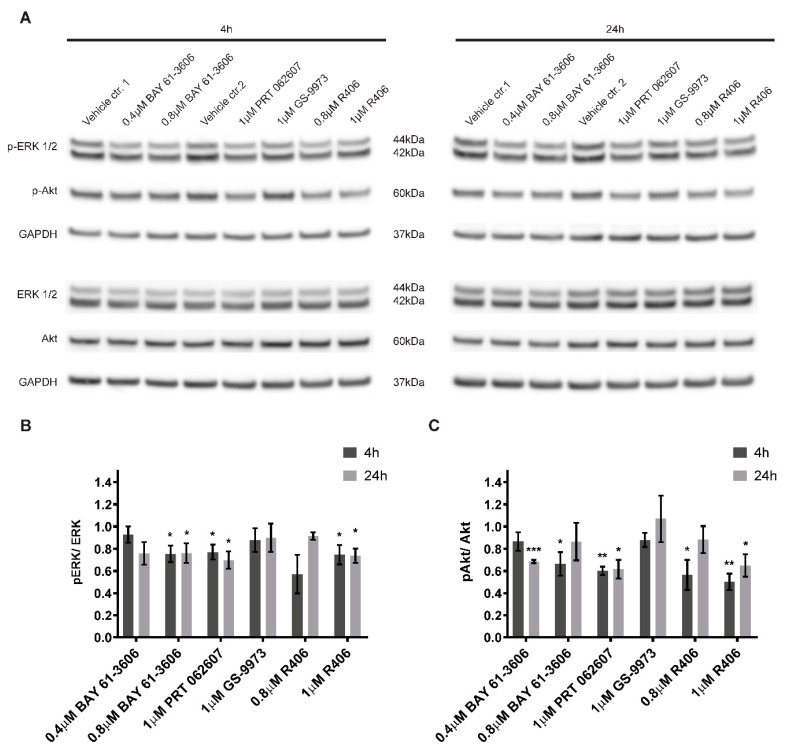
SYK inhibition decreases ERK1/2 and Akt phosphorylation. SYK inhibitors BAY 61-3606, R406, and PRT062607 reduce the phosphorylation of ERK1/2 and Akt in SH-SY5Y cells after a 4 or 24 h treatment (**A**). Control cells were treated with the corresponding vehicles (water = vehicle control 1 and DMSO = vehicle control 2). Densitometric analysis of the protein bands was performed. Phosphorylated and total protein were normalized to their respective GAPDH loading controls and the ratios between normalized p-ERK1/2 and ERK as well as normalized p-Akt and Akt were calculated. The values are displayed as mean ± SD relative to the vehicle control = 1 (**B** and **C**). The results are based on three independent experiments. One sample t-test (two-tailed) was performed to compare the vehicle (theoretical mean = 1) vs. inhibitor treatment: * *p* < 0.05 ** *p* < 0.01 *** *p* < 0.001.

**Figure 7 cancers-11-00202-f007:**
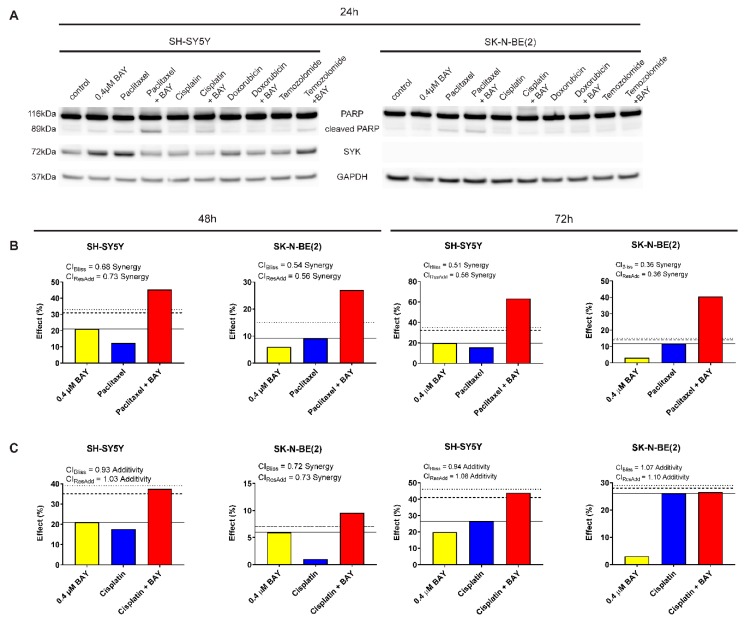
The selective SYK inhibitor BAY 61-3606 enhances the effect of chemotherapeutic drugs in neuroblastoma cells. (**A**) PARP cleavage and SYK expression were determined by western blot after 24 h monotherapy or combinations of 0.4 µM BAY 61-3606, 20 nM paclitaxel, 5 nM doxorubicin, 100 µM temozolomide and cisplatin (1 µM or 3 µM for SH-SY5Y and SK-N-BE(2), respectively). Illustration of drug combination effects for 0.4 µM BAY 61-3606 and paclitaxel (**B**) as well as 0.4 µM BAY 61-3606 and cisplatin (**C**) in SH-SY5Y and SK-N-BE(2) cells after 48 h and 72 h treatment. The continuous horizontal line indicates the effect of the highest single agent, the dashed line denotes expected additive effect calculated by the Bliss independence model, and the dotted line shows expected additive effect calculated by response additivity. Combination index (CI), given from the Bliss independence model and the response additivity, and effect are specified for each combination.

**Figure 8 cancers-11-00202-f008:**
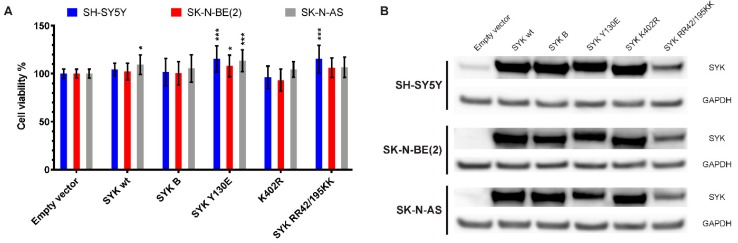
Transfection with an active SYK variant increases the cell viability of neuroblastoma cells independent of endogenous SYK levels. Cells were transfected with expression plasmids encoding different SYK variants and the cell viability was measured after 48 h by MTT assay (**A**). The empty vector control was set as 100% viable cells. Data are presented as mean ± SD from six independent experiments. Statistical comparisons were made using two-way ANOVA and a difference was observed between the SYK variants and between cell lines *p* < 0.001. The Dunnett’s multiple comparison test was used to evaluate the difference between empty vector and different SYK variants: * *p* < 0.05, *** *p* <0.001. Western blot analysis was used to evaluate the presence of SYK following transfection, ensuring sufficient transfection efficiency (**B**). SYK wt = SYK wild type, SYK B = short SYK splice variant B, SYK Y130E = constitutive active SYK, SYK K402R = kinase dead SYK, SYK RR42/195KK = SYK with inactive SH2 domains.

**Table 1 cancers-11-00202-t001:** Presence of SYK and p-SYK (Tyr525) in neuroblastoma tumor tissue.

Neuroblastoma Subgroups	SYK Positive (Sections Examined)	p-SYK Positive (Sections Examined)
Neuroblastoma	40 (42)	38 (40)
Non-*MYCN*-amplified	31 (32)	29 (31)
*MYCN* amplified	9 (10)	9 (9)
* Treated tissue	11 (13)	10 (11)
* Untreated tissue	26 (26)	25 (26)
Ganglioneuroma	3 (3)	3 (3)

* For three tumor tissue samples the information concerning prior treatment was unavailable.

**Table 2 cancers-11-00202-t002:** Cell viability of SH-SY5Y and SK-N-BE(2) after treatment with 0.4 μM BAY 61-3606, chemotherapeutic drugs or combinations of both.

		SH-SY5Y			SK-N-BE(2)		
	Treatment	Cell viability (%) Mean ± SD	*p* value Drug vs. combination	*p* value BAY vs. combination	Cell viability (%) Mean ± SD	*p* value Drug vs. combination	*p* value BAY vs. combination
**48 h**	0.4 μM BAY	79.01 ± 6.26			94.04 ± 7.72		
	Paclitaxel	87.71 ± 7.83			90.89 ± 7.86		
	Paclitaxel + BAY	54.65 ± 3.26	**<0.001**	**<0.001**	72.98 ± 9.33	**<0.001**	**<0.001**
	Cisplatin	82.33 ± 9.01			98.94 ± 6.48		
	Cisplatin + BAY	62.47 ± 5.82	**<0.001**	**<0.001**	90.49 ± 7.46	0.052	>0.999
	Doxorubicin	100.5 ± 9.59			102 ± 9.68		
	Doxorubicin + BAY	77.97 ± 8.01	**<0.001**	>0.999	93.69 ± 8.78	0.121	>0.999
	Temozolomide	107 ± 11.17			107.7 ± 3.65		
	Temozolomide + BAY	76.81 ± 4.82	**<0.001**	>0.999	100 ± 4.36	0.132	0.181
**72 h**	0.4 μM BAY	80.20 ± 7.62			96.97 ± 6.89		
	Paclitaxel	84.53 ± 4.60			88.29 ± 5.19		
	Paclitaxel + BAY	36.79 ± 3.14	**<0.001**	**<0.001**	59.48 ± 8.81	**<0.001**	**<0.001**
	Cisplatin	73.38 ± 5.89			73.99 ± 2.95		
	Cisplatin + BAY	56.25 ± 5.72	**<0.001**	**<0.001**	73.53 ± 5.94	>0.999	**<0.001**
	Doxorubicin	102.2 ± 8.37			101.3 ± 7.56		
	Doxorubicin + BAY	78.57 ± 7.16	**<0.001**	>0.999	93.99 ± 9.68	0.053	>0.999
	Temozolomide	107.7 ± 6.13			102.7 ± 4.28		
	Temozolomide + BAY	74.21 ± 5.99	**<0.001**	0.217	99.74 ± 6.68	>0.999	>0.999

Cell viability was measured by MTT assay after 48 and 72 h. The control was set as 100% viable cells. Data are presented as mean ± SD from at least three independent experiments. Using two-way ANOVA, a significant effect was observed for both treatment and between cell lines *p* < 0.001; Bonferroni’s multiple comparison test was used to evaluate differences between treatments, and *p* values < 0.05 were considered as statistically significant.

**Table 3 cancers-11-00202-t003:** Antibodies.

Antibody	Application	Source
Anti-SYK	WB, IP	#1240, Santa Cruz Biotechnology
Anti-SYK	WB	#13198, Cell Signaling Technology
Anti-SYK	ICC/IHC	#HPA001384, Sigma
Anti-Phospho-ZAP-70 (Tyr319)/SYK (Tyr352)	WB	#2701, Cell Signaling Technology
Anti-Phospho-SYK (Tyr323)	WB	#2715, Cell Signaling Technology
Anti-Phospho-SYK (Tyr525/526)	ICC	#2710, Cell Signaling Technology
Anti-Phospho-SYK (pTyr525)	IHC	#SAB4503839, Sigma
Anti-PARP	WB	#9542, Cell Signaling Technology
Anti-Phospho-p44/42 MAPK (Erk1/2) (Thr202/Tyr204)	WB	#4370, Cell Signaling Technology
Anti-p44/42 MAPK (Erk1/2)	WB	#4695, Cell Signaling Technology
Anti-Phospho-Akt (Ser473) (D9E)	WB	#4060, Cell Signaling Technology
Anti-Akt	WB	#9272, Cell Signaling Technology
Anti-GAPDH	WB	#47724, Santa Cruz Biotechnology
Goat Anti-Rabbit IgG H&L (HRP)	WB	#6721, Abcam
Rabbit Anti-Mouse IgG H&L (HRP)	WB	#97046, Abcam
Goat anti-Rabbit IgG (H+L), Alexa Fluor 488	ICC	# A-11008, Thermo Fisher Scientific

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
