# Peer review of "SYK Inhibition Potentiates the Effect of Chemotherapeutic Drugs on Neuroblastoma Cells In Vitro"

_cancers, 2019, doi:10.3390/cancers11020202_

Round 1
Reviewer 1 Report
Overall assessment for the manuscript titled “SYK inhibition potentiates the effect of chemotherapeutic drugs on neuroblastoma cells in vitro”. In this study, the authors examined the expression and activity of SYK in neuroblastoma tumor tissues and established cell lines, and investigated its potential as a therapeutic target. I find this very novel given that there are few reports available for this target in neuroblastoma, although the results presented lead to many questions to be answered. I do recommend this study for acceptance after the following concerns being clearly addressed by the authors.
1. In Table 1, could the authors provide information for these tumors in regards to whether they were post-chemotherapy cancer tissues? And whether and which were treated with any of the chemotherapy agents tested in vitro in this study. This curiosity is due to prior reports showing a role for SYK upregulation/activation in paclitaxel-treated cancers, and it could serve as biomarker for cisplatin efficacy. This might, to some extent, explain the discrepancy between SYK levels in tumor tissues and established cell lines.
2. In Fig. 4 and Supplemental Fig. 4 and 5, were the SYK inhibitors and chemotherapy drugs administrated simultaneously in this study? Would sequential treatment, especially if SYK inhibitor being added later, provide a stronger efficacy? This is also based on the prior mechanistic studies for roles of SYK in chemotherapy resistance.
3. In Fig. 5, BAY 61-3606 displayed strongest phenotype among all tested SYK inhibitors. And BAY was subsequently selected for all downstream experiments. Are similar observations in these experiments also reproduced by other SYK inhibitors, although to less extent? Does this trend correlate with their reported activity and SYK selectivity?
4. In Fig. 6, please provide the western blotting of total AKT and ERK1/2, to eliminate the possibility of a decrease of the activity (phosphorylation) of these markers due to a decrease of total levels after treatments.
5. In Fig. 8, the SYK-expressing plasmids were transfected into neuroblastoma cells expressing WT SYK, to assess the effects on SYK on ‘cell viability’. It should be assessing the proliferation of cells with higher levels of SYK proteins. Was SYK Knockdown, as in Fig. 4, performed before ectopic overexpression? If not, the constitutive expression of SYK, especially that in SK-SY5Y, may prevent a clear conclusion.
Author Response
Dear Reviewer #1,
Thank you for highly relevant comments and suggestions.
We have now carefully revised the manuscript and hope we have provided adequate answers to your comments.
In the attached pdf file we provide a point-by-point response to your comments (in italics) in detail.
On behalf of all co-authors.
Conny Tümmler

Reviewer 2 Report
Review of:
Manuscript ID: cancers-419340
Type of manuscript: Article
Title: SYK inhibition potentiates the effect of chemotherapeutic drugs on
neuroblastoma cells in vitro
Authors: Conny Tümmler *, Gianina Dumitriu, Malin Wickström, Peter Coopman,
Andrey Valkov, Per Kogner, John Inge Johnsen, Ugo Moens, Baldur
Sveinbjörnsson
Summary
This manuscript makes a case that SYK plays a role in neuroblastoma and that inhibiting SYK may provide benefit to patients in combination with other drugs. The case for a role of SYK in neuroblastoma is built based on higher SYK gene expression in some cohorts vs. neural crest; and IHC staining in tumors, although expression in cell lines based on RT-PCR and western blotting is less frequent. Inhibiting SYK by siRNA and drugs had modest affects on neurblastoma cell lines, in some cases whether they expressed SYK or not. Drugs that inhibit SYK when used in combination with other drugs showed some additive and synergistic affects on PARP cleavage and growth of neuroblastoma cell lines, again in some cases independent of SYK expression.
General comments
Although many of the results appear to be statistically significant as presented, the case that SYK is a potential therapeutic target for treatment of neuroblastoma is not strong. My concerns can be grouped into two categories, A) methodological shortcomings, and B) biological significance. Specific comments about methods follow. The general concern about biological significance is that the biological effects of manipulating SYK activity or expression are small, 10-20% at best, and the drug affects are independent of SYK expression.
Methodological shortcomings
Cohort is defined as a group based on one or more defined characteristics. How are the cohorts identified for the gene expression analysis in Fig 1A? The y-axis should be explained in this figure (the graph header seems out of place for a publication). None of the cohorts are very far above zero, which does not suggest evidence for high gene expression. What was used as neural crest tissue?
IHC is used to screen tumor samples from patients with selected examples shown in Figure 1B and negative examples in supplementary Fig 2. It appears that this was the only criteria for positive SYK expression in Table 1. Western blots should be used to confirm these data.
The contrast between the number of SYK-positive tumors vs. neuroblastoma cell lines also raises the concern that IHC dramatically overestimates expression. Also, there are inconsistencies with the gene expression data (Supp. Fig 1) and MYC vs. MYCN expression; the ‘nuclear’ staining, which is not seen in Fig. 2C and D; and there is concern about background in Fig 2E in SK-N-BE(2) cells that have no SYK mRNA or protein. The latter half of the statement in line 142, “This could be attributed to some moderate non-specific binding of the antibody or very low levels of SYK,” does not make sense - do the investigators not believe their data showing that SK-N-BE(2) cells have no SYK mRNA or protein? Further concerns are raised by comparing staining in Figure 3B to that in 2E: they look very similar except for one or two puncta (which could be background), yet staying in 3E is deemed positive and 2E negative.
MTT assays are used throughout the paper to estimate cell viability. It is now well established that comparisons of cell growth rate in response to perturbagens is highly variable and unreliable, but there is now a method to normalize these data that is much more robust (Hafner, M., Niepel, M., Chung M., Sorger P.K. (2016), “Growth rate inhibition metrics correct for confounders in measuring sensitivity to cancer drugs”. Nature Methods, 13(6), 521- 527.) The GRI method should be used to calculate effects in Figures 4A; 5 (all); 6B & C; Table 2; and 8A (and corresponding supplementary figures). There are other better methods to measure cell viability (e.g. “live/dead” assay; cleaved-caspase-specific assay).
It appears that PARP cleavage (an assay for apoptosis) was used to determine additive or synergistic effects of drugs. The total amount of PARP cleaved appeared to be small, maybe 10% of the total, which is in line with the small effects seen with the MTT assays. It is not clear that PARP cleavage is the appropriate assay for the Bliss independence model. Furthermore, this model is not rigorous or statistically robust without a 2-stage surface model that tests drugs at different concentrations (see Zhoa, et al., 2014 J Biomed Screen 19(5):817-21 and Foucquier and Guedj, 2015, Pharm Res Persp e00149). The investigators should test for additive and synergistic effects using more robust methods according to these papers. There are other methods to measure apoptosis that should be brought to bear (as mentioned above).
Biological significance
The effect sizes are rather small in Figure 4E (siRNA), 5 (SYK inhibitors); Table 2, Fig 8, etc. If data are expressed as normalized growth rate ratios (GRI) as suggested above the effect size may be more believable but not likely to get larger. Similarly, the effect sizes in Figure 6 (ERK and AKT phosphorylation) are not impressive, and there is no dose-dependence for BA 61-3606. These results are not likely to motivate further experiments with an eye towards clinical applications.
Effects of SYK inhibitors on SK-N-BE(2) cells that have no SYK mRNA or protein must surely be off-target effects. We need to see the data in Figure 5 expressed as GRI as noted above, but in any case the concentration of drugs that affect SH-SY5Y cells and not SK-N-BE(2) cells may be due to other reasons. SK-N-BE(2) cells also lack p53 and are exceptionally vigorous growers. It is certainly not consistent with the conclusion that SYK plays a role in neuroblastoma when drug effects, even a synergistic effect between the BAY drug and taxol, was observed in cells that lack SYK. The effects of manipulating SYK expression on cell growth were modest, at best, and are not sufficient in themselves to be convincing.
Author Response
Dear Reviewer #2,
Thank you for highly relevant comments and suggestions.
We have now carefully revised the manuscript and hope we have provided adequate answers to your comments.
In the attached pdf file we provide a point-by-point response to your comments (in italics) in detail.
On behalf of all co-authors.
Sincerely yours,
Conny Tümmler

Round 2
Reviewer 1 Report
After this round of revision, I believe the authors have addressed most
of my concerns. Therefore, I now recommend it for acceptance as it is. Thank you.
Author Response
Dear Reviewer #1,
Thank you very much for your excellent review and the resulting improvements of the manuscript.
On behalf of all co-authors.
Sincerely yours,
Conny Tümmler
Reviewer 2 Report
Overall, the text revisions have sufficiently improved the clarity of
the experimental procedures, analysis, and sample information.
However, we have two main concerns that have not been sufficiently
addressed following the authors’ revisions.
First, it remains unclear why there is differential staining between
Figures 1, 2 and 3.
In their response, the authors state: “SYK is a non-receptor tyrosine
kinase localized in the cytoplasm and nucleus [15]. Upon
phosphorylation, an increased localization of SYK to the nucleus and
cytoplasm has previously been described [16]. Therefore, labeling with a
phospho-specific antibody as seen in Figure 1D and E can show a more
pronounced nuclear staining in comparison to labeling with an antibody
that detects total SYK where both nuclear and cytoplasmic staining was
observed.”
In this case, the antibody that recognizes total SYK should exhibit both
nuclear and cytoplasmic staining in Figure 2 (recognizing both the
active and inactive form) - however, staining appears to be cytoplasmic.
Furthermore, the pSYK antibody used in Figure 3 does not exhibit nuclear
staining at all, which contradicts the IHC data. The concern is that
there may be artifactual staining in the IHC data that overestimates the
number of SYK-positive cells.
Second, the authors did not improve the analysis of the MTT/cell
viability data. We suggested calculating normalized growth rate
inhibition values (GR) to account for changes in division time for
different cell lines throughout the experiments. The concern is if
control cells undergo a change in growth rate during the course of the
experiment, then comparing cell numbers at discrete time points can be
misleading. For example, in faster growing cell lines compared to slower
growing cell lines, cell number can increase dramatically while the
actual division time decreases over the course of the experiment. The
reason that this is important is that the effects are small and it has
been rigorously shown that without normalization there is a danger of
artifact. The authors point out that small effects can lead to progress,
agreed, but they have to be real, and the GR calculation has been shown
to be rigorous and thus more convincing.
To make this easier there is an online calculator
http://www.grcalculator.org/grcalculator/. See
http://www.grcalculator.org/grtutorial/Home.html. “Normalized growth
rate inhibition (GR) values are based on the ratio of growth rates in
the presence and absence of perturbagen. Largely independent of cell
division rate and assay duration, GR metrics are more robust than IC50
and Emax for assessing cellular response to drugs, RNAi, and other
perturbations in which control cells divide over the course of the
assay.”
Author Response
Dear Reviewer #2,
Thank you for the highly relevant comments.
We have now carefully considered your comments and hope we have provided adequate answers.
Please find our detailed point-by-point response in the attached pdf file.
On behalf of all co-authors.
Sincerely yours,
Conny Tümmler
